Effects of core stability exercises on balance among Chinese children and youth with intellectual disabilities

Qi Jing 1
Zhong Yecheng 1
Zhou Junjie 1
Li Niuniu 1
Han Yuhang 1
Dong Zehua 1
Li Yan 2
Xu Wenhong xuwenhong@zjnu.edu.cn 1
1 College of Physical Education and Health Sciences, Zhejiang Normal University , Jinhua , Zhejiang Province , China
2 Zhejiang Yuying College of Vocational Technology , Hangzhou , Zhejiang Province , China
Farhan Faiza
Electronic publication date: 2025 Dec 9
Publication date: 2025
Volume: 13
Electronic Location ID: e20454
Received 2025 Mar 21; Accepted 2025 Oct 31
Copyright: ©2025 Qi et al.
Copyright year: 2025
Copyright holder: Qi et al.
License: This is an open access article distributed under the terms of the Creative Commons Attribution License, which permits unrestricted use, distribution, reproduction and adaptation in any medium and for any purpose provided that it is properly attributed. For attribution, the original author(s), title, publication source (PeerJ) and either DOI or URL of the article must be cited.
License URL: https://creativecommons.org/licenses/by/4.0/

Keywords: Balance, Core stability exercises, Intellectual disabilities

Funding: The Project of Education on Common Prosperity of Zhejiang Normal University, Zhejiang Province, China No. 24JYGF27 This research was supported by the Project of Education on Common Prosperity of Zhejiang Normal University (No. 24JYGF27), Zhejiang Province, China. The funders had no role in study design, data collection and analysis, decision to publish, or preparation of the manuscript.

==============================
Background

Children and youth with intellectual disabilities (IDs) are at risk of falls due to balance problems.

Objective

This study was conducted to assess the effects of six weeks of core stability exercises on balance in this population.

Methods

Thirty-five participants with IDs from a special school were selected. Static and dynamic balance were assessed by the one-legged stance test (OLS) and lower quarter Y balance test (YBT-LQ), respectively.

Results

Core stability exercises significantly improved static balance with eyes opened (EO) on firm (FI) ground (t = −5.269, P < 0.001) and dynamic balance in all three directions (left leg: anterior: t = −3.197, P = 0.005; posteromedial: t =  − 4.608, P < 0.001; posterolateral: t = −4.706, P < 0.001; right leg: anterior: t = −3.324, P = 0.004; posteromedial: t = −5.614, P < 0.001; posterolateral: t = −5.905, P < 0.001).

Conclusions

Six weeks of core stability exercises significantly improved the static balance of children and youth with IDs under EO and FI conditions, as well as all three directions of dynamic balance. Future studies with multiple measurement points are warranted to examine the long-term effectiveness of this training on balance in this population.

Introduction

Children and youth with intellectual disabilities (IDs) generally exhibit worse balance performance than their typically developing peers do (Kavanagh, Manninen & Issartel, 2023; Sretenović, Nedović & Ð orđević, 2019). Maintaining balance involves the coordinated operation and integration of the sensory system, the central nervous system and the musculoskeletal system (Jouira, Rebai & Sahli, 2023; Paillard, 2017). The sensory input system provides the body with information about its relative position in the surrounding environment. This information is then processed by the central control system and sent to effectors (such as joint muscles), which respond according to the received information, thereby controlling or altering the body’s posture (Zhao et al., 2024; Cao et al., 2021). However, children and youth with IDs often suffer from sensory deficits (visual, proprioceptive and vestibular deficits), cognitive impairments, and delayed development of joint and muscle functions (Blomqvist et al., 2013; Leyssens et al., 2022); this makes them more prone to balance-related problems. In addition, the discrimination and stigmatization against these children, along with their low self-efficacy, decrease their motivation to participate in sports activities (Puce et al., 2023). Thus, limited exercise experience and a lack of accessible training programs may also lead to motor dysfunction in children and youth with ID, reducing their balance ability (Ghaeeni, Bahari & Khazaei, 2015).

Insufficient balance ability not only increases the risk of these children falling and suffering from fall-related injuries but also leads to low self-esteem levels and a sedentary lifestyle (Paillard, 2017; Lee & Jeoung, 2016). Frequent prolonged sedentary behavior, in turn, further damages motor function, which can ultimately lead to a decline in independence and a decrease in quality of life. Given the significance of balance for the daily activities of children and youth with IDs, researchers have conducted many studies in this field. For example, Jouira et al. (2024) investigated the factors influencing the postural balance of adolescents with IDs. They reported that in a dual-task situation, cognitive tasks impair postural balance and emphasised the positive impact of physical activity on balance ability. In addition, core stability exercises have become common elements of training programs aimed at enhancing the balance performance of these children (Ahmadi, Hasan & Hosin, 2012; Alsakhawi & Elshafey, 2019). The theory of motor learning emphasises that a series of practice or experience processes can lead to relatively permanent changes in motor ability, which provides a solid theoretical foundation for this research paradigm (Arntzen et al., 2021). The core muscles are located around the body’s center of gravity in the anatomical position. Core stability training helps to strengthen core muscle groups (such as the multifidus, transverse abdominis, and pelvic floor) and thereby significantly influences a person’s posture by stabilizing the pelvis and controlling posture fluctuations (Arnold et al., 2015; Sadeghi et al., 2020).

Based on quantitative evidence, a systematic review and meta-analysis by Zhou, Zhong & Xu (2024) summarised the effectiveness of core stability exercises in improving balance among children and youth with IDs; their findings revealed that compared with active control groups, core stability exercises improved participants’ dynamic balance but had no significant effect on static and static–dynamic balance; however, given the scarcity of studies included in their review, the findings cannot be used to draw definitive conclusions. Therefore, the aim of this study was to assess the effect of six weeks of core stability exercises on balance among children and youth with IDs. We hypothesised that core stability exercises may improve the balance ability of these children. The findings of our study can assist medical staff, school teachers, and caregivers in implementing targeted interventions to increase the balance ability of individuals with IDs, thereby improving their overall quality of life.

Methods

Study design

This study was a single-center, two-armed, evaluator-blinded randomised controlled trial (RCT). The study protocol was registered on the website of the Chinese Clinical Trial Registry (Registration no. ChiCTR2400088510). Ethical approval was obtained (ZSRT2024189) from the Ethics Committee of Zhejiang Normal University. The study was reported in accordance with the Consolidated Standards of Reporting Trials (CONSORT) statement for randomised trials (Schulz et al., 2010).

Participants and setting

Children and youth with IDs were recruited from a special school in a city in southeastern China. This school includes primary school (grades 1–6, aged 6–12 years) and middle school (grades 7–9, aged more than 13 years) and provides educational services for students with various types of disabilities.

Initially, each class teacher recommended potentially eligible students for the researchers according to the following criteria: (1) presence and severity of ID were diagnosed by a physician based on the fifth edition of the Diagnostic and Statistical Manual of Mental Disorders (American Psychiatric Association, 2016); (2) aged 6–18 years old; (3) able to follow instructions and communicate verbally; and (4) had no history of core stability, strength or balance training during the past six months. The exclusion criteria included (1) an uncontrolled cardiovascular disorder; (2) neurological disorders (e.g., epilepsy); (3) vision or hearing impairments that hinder exercise performance; and (4) musculoskeletal limitations, such as spinal, hip, knee or foot deformities. Among the 53 individuals who consented to participate, 15 were not included because of the exclusion criteria (Fig. 1).

Figure 1 The CONSORT flowchart of the study and the process of the allocation of subjects to research groups.

Randomisation and masking

After completing the baseline assessment, the participants were randomly assigned (1:1) to receive either core stability exercises (intervention group) or their regular school schedule (control group). Allocation was based on a computer-generated random number list prepared by a researcher with no involvement in the trial. Specifically, a sealed envelope contained a slip of paper with a random number indicating the group assigned to each subject after baseline assessments.

As in most trials, it was not possible to “blind” the subjects regarding the intervention options. In the present study, only the assessor was blinded; therefore, the study did not involve unblinding.

Intervention and control

Before the trial began, all assessment and intervention personnel received uniform training and developed common operating standards. Intervention and evaluation procedures were carried out in strict accordance with standardised methods.

The participants in the intervention group received Jeffreys’ core stability exercises (Jeffreys, 2002; Aly & Abonour, 2016; Ghaeeni, Bahari & Khazaei, 2015; Alsakhawi & Elshafey, 2019), which included lumbar–pelvic proprioception retraining, specific spinal stabilisation exercises, and different muscle contractions and abdominal maneuvers (Alsakhawi & Elshafey, 2019). In particular, this protocol consists of three progressive levels: level 1 is static contraction training conducted under stable conditions; level 2 is dynamic training, which is also performed under stable conditions; and level 3 includes dynamic and resistance training, which is performed under unstable conditions (such as with Swiss balls). The specific contents and methods of this training are shown in Table 1.

Table 1 Jeffreys’ core stability exercises (Alsakhawi & Elshafey, 2019; Jeffreys, 2002).

Week	Form of exercise	The volume and intensity of exercise	
	Contracting abdominal muscles while lying in a supine position	Three sets and 20 repetitions in each set	
1 and 2	Contracting abdominal muscles while lying in a prone position	Three sets and 20 repetitions in each set	
	Contracting abdominal muscles while in a squat position	Three sets and 20 repetitions in each set	
	Contracting abdominal muscles while lying in a supine position with one leg stretched and the other bent at the knee and pressed against the abdomen	Three sets and 20 repetitions in each set	
3	Contracting abdominal muscles while lying in a prone position with one leg stretched and the body weight on the other leg which is bent at the knee	Three sets and 20 repetitions in each set	
	Side lying bridge for each side of the body	Six repetitions, a 10-s pause	
	Contracting abdominal muscles while lying in a supine position and pulling the limbs upward with arms and legs kept close	Three sets and 20 repetitions in each set	
4	In squat position, one leg is raised and pulled outward and backward	Three sets for each leg and 20 repetitions in each set	
	Trunk rotation while holding weights in each hand	Three sets each part of the body and 20 repetitions in each set	
	Sitting on a Swiss ball and holding the abdomen in	Three sets, 10 s	
5	Squatting while the Swiss ball is on the shoulder	Three sets and 15 repetitions for each set	
	Bringing up the arms and legs simultaneously in the prone position	Three sets and 15 repetitions for each set	
	Bending 45° to the left or right	Three sets for each side, 12 repetitions in each set	
6	Bridging while shoulders and hands are on the floor and one leg is raised	Three sets for each leg, 15 repetitions in each set	
	Contracting abdominal muscles while lying in a supine position on the Swiss ball	Three sets, 12 repetitions in each set	

Participants in the intervention group received training three times a week, 35 min per session, for a total of 6 weeks. They were also encouraged not to seek any physical therapy other than core exercises during the intervention. Participants in the control group continued their regular school schedule (i.e., 35 min of PE lessons, three times per week) and were encouraged to maintain their current level of physical activity.

Outcome measurements

Two assessors who were masked with respect to group allocation conducted the assessments at baseline and at week 7. The height (cm) and weight (kg) of the participants in both groups were measured using a portable instrument (Inbody BSM370, Korea). Height was measured to the nearest 0.10 cm in bare feet, and weight was measured to the nearest 0.10 kg. Body mass index (BMI; kg/m2) was subsequently calculated.

Assessment of static balance

One-legged stance (OLS) test was used to evaluate static balance. The participants were instructed to stand on the dominant leg and place the heel of the other (free) leg on the knee of the supporting leg. Hands were placed on both sides of their hip. The assessment was conducted under four conditions: (1) standing with eyes opened on firm ground (EO, FI); (2) standing with eyes closed on firm ground (EC, FI); (3) standing with eyes opened on a foam pad (i.e., AIREX balance pad with 40  × 50 cm dimensions and six cm thickness; EO, FO); and standing with eyes closed on a foam pad (EC, FO). Under these four conditions, the participants were encouraged to stand for as long as possible to 60 s. The test is completed when the participant’s free leg is placed on the floor. If they opened their eyes during a test requiring closed eyes, they were required to take the test again. The assessors used a chronometer to record the participants’ standing time in seconds. The best of three attempts was recorded. The OLS test has been applied to children and youth with IDs (Rahmat & Hasan, 2013; Yılmaz et al., 2009) and has demonstrated good reliability and validity (Blomqvist et al., 2012).

Assessment of dynamic balance

A lower quarter Y balance test (YBT-LQ) kit (Functional Movement Systems, Chatham, VA, USA) was used to measure the participants’ dynamic balance performance. The test is a valid and reliable measure of dynamic balance in children and youth with IDs (Ahmadi, Hasan & Hosin, 2012; Balayi, Sedaghati & Ahmadabadi, 2022; Jouira, Rebai & Sahli, 2023). The test kit consisted of a centralised stance platform to which three pipes were attached to represent the anterior (AT), posteromedial (PM) and posterolateral (PL) reach directions. Each pipe was marked in 1.0 cm increments for measurement purposes and was equipped with a moveable reach indicator.

Each participant was asked during the YBT-LQ to stand barefoot (to eliminate potential effects of varying footwear). One foot was on the center foot plate, and the most distal aspect of the toes was behind the starting line (Gribble, Hertel & Plisky, 2012). While maintaining a single-leg stance, each participant was asked to push the reach indicator with the reaching foot as far as possible and return to the original start position while maintaining balance. The test direction order was standardised as follows: right AT, right PM, right PL, left AT, left PM and left PL (Jouira, Rebai & Sahli, 2023). Each participant performed three practice trials (unaccounted tests) for each direction so that participants became comfortable with performing the task. After two minutes of rest, each participant performed three test trials in each direction (accounted tests). To avoid the potential effects of fatigue and learning, a 60 s rest period was provided between trials, during which participants were allowed to sit down. A trial was classified as invalid when the participants (1) lost their balance (i.e., stepped with the reach leg on the ground), (2) lifted the stance leg from the stance platform, (3) stepped on top of the reach indicator for support or (4) kicked the reach indicator (Muehlbauer & Waldermann, 2022; Plisky et al., 2009). When an invalid trial occurred, the corresponding data were discarded, and trials were repeated until a total of three valid trials were completed. The maximal reach distance (cm) per reach direction was used for further analysis.

Statistical analyses

The sample size was determined in advance using G*power software (version 3.1.9.2). Based on a confidence interval of 95%, a power of 80% and an attrition rate of 10%, 19 participants were needed per group, yielding a total sample size of 38.

Data were analysed by researchers using SPSS 27.0 (IBM Corp., Armonk, NY, USA). The normal distribution of the data was analysed by the Shapiro–Wilk test and histogram graphs. Descriptive analyses are presented as the mean and standard deviation for continuous variables and numbers and percentages for categorical variables. Prior to conducting independent t tests, Levene’s test for equality of variances was performed. The baseline demographic characteristics of categorical and continuous variables were compared between groups using chi-square tests and independent t tests, respectively. Differences within each group before and after the intervention were analysed using paired t tests. The differences between the groups before and after the intervention were analysed via independent t tests. Statistical significance was set at P < 0.05 for all tests.

Results

Comparison of static and dynamic balance between groups before and after the intervention

The demographic characteristics of the participants in both groups are presented in Table 2. The intervention and control groups were homogenous in terms of all the demographic characteristics (P > 0.05).

The results in Table 3 show no statistically significant difference between the intervention and control groups in terms of static and dynamic balance before the intervention (P > 0.05).

After the intervention, independent t tests (Table 4) revealed statistically significant differences between the two groups in one stance condition (i.e., OLS time; EO, FI) of static balance (t = −3.105, P = 0.004) and in all directions for both legs of the dynamic balance (left leg: AT direction: t = −4.510, P < 0.001; PM direction: t = −4.241, P < 0.001; PL direction: t = −4.403, P < 0.001; right leg: AT direction: t = −4.920, P = 0.004; PM direction: t = −4.821, P < 0.001; PL direction: t = −4.963, P < 0.001).

Effects on static balance performance

Pre- and post-comparisons within the groups revealed significant differences in the intervention group in only one stance condition (i.e., OLS time; EO, FI; t = −5.269, P < 0.001) but not in the control group in all four stance conditions (Fig. 2).

Table 2 Demographic characteristics of the participants.

	M ± SD	
	Intervention group (n = 18)	Control Group (n = 17)	P-value	
Age, years	11.83 ± 1.25	12.18 ± 1.19	0.411	
Gender, n (%)				
Boys	11 (61.11)	10 (58.82)	0.089	
Girls	7 (38.89)	7 (41.18)	
Height, m	1.45 ± 0.10	1.46 ± 0.08	0.689	
Weight, kg	41.73 ± 8.94	38.32 ± 8.77	0.263	
BMI, kg/m2	20.01 ± 4.55	17.87 ± 3.42	0.126	
ID severity, n (%)				
Moderate	12 (66.67)	13 (76.47)	0.521	
Severe	6 (33.33)	4 (23.53)	
Notes.

M, mean values; SD, standard deviations; BMI, Body Mass Index; ID, intellectual disability.

Table 3 Comparison of static and dynamic balance between groups before the intervention.

Parameter	M ± SD	t	P-value	
	Intervention group (n = 18)	Control group (n = 17)			
Static balance					
OLS time; EO, FI (sec)	7.61 ± 4.30	6.81 ± 3.49	−0.603	0.551	
OLS time; EC, FI (sec)	4.40 ± 2.71	4.00 ± 2.35	−0.475	0.638	
OLS time; EO, FO (sec)	3.72 ± 2.07	3.03 ± 1.84	−1.046	0.303	
OLS time; EC, FO (sec)	2.07 ± 0.94	1.87 ± 0.87	−0.796	0.432	
Dynamic balance					
Left					
AT (cm)	41.44 ± 15.72	42.06 ± 9.23	0.140	0.890	
PM (cm)	57.50 ± 16.62	56.76 ± 12.11	−0.149	0.883	
PL (cm)	61.17 ± 19.26	62.12 ± 11.47	0.176	0.861	
Right					
AT (cm)	41.89 ± 16.53	41.94 ± 11.78	0.011	0.992	
PM (cm)	58.61 ± 15.40	56.59 ± 15.10	−0.392	0.697	
PL (cm)	62.17 ± 15.43	65.06 ± 15.04	0.561	0.579	
Notes.

M, mean values; SD, standard deviations; OLS, one-legged stance; EO, eyes opened; EC, eyes closed; FI, firm ground; FO, foam ground; AT, anterior; PM, posteromedial; PL, posterolateral.

Table 4 Comparison of static and dynamic balance between groups after the intervention.

Parameter	M ± SD	t	P-value	
	Intervention group (n = 18)	Control group (n = 17)			
Static balance					
OLS time; EO, FI (sec)	12.61 ± 7.40	6.41 ± 3.70	−3.105	0.004	
OLS time; EC, FI (sec)	4.76 ± 2.81	4.06 ± 2.65	−0.758	0.454	
OLS time; EO, FO (sec)	4.12 ± 2.04	3.50 ± 1.96	−0.918	0.366	
OLS time; EC, FO (sec)	2.46 ± 1.09	2.07 ± 1.16	−1.016	0.317	
Dynamic balance					
Left					
AT (cm)	55.94 ± 11.87	40.53 ± 7.80	−4.510	<0.001	
PM (cm)	79.39 ± 16.30	59.41 ± 10.86	−4.241	<0.001	
PL (cm)	85.67 ± 15.61	64.18 ± 13.05	−4.403	<0.001	
Right					
AT (cm)	57.22 ± 12.61	39.65 ± 8.16	−4.920	<0.001	
PM (cm)	80.94 ± 15.19	59.18 ± 11.07	−4.821	<0.001	
PL (cm)	90.11 ± 16.53	64.24 ± 14.14	−4.963	<0.001	
Notes.

M, mean values; SD, standard deviations; OLS, one-legged stance; EO, eyes opened; EC, eyes closed; FI, firm ground; FO, foam ground; AT, anterior; PM, posteromedial; PL, posterolateral.

Figure 2 (A–B) Comparison of static balance performance of the participants within the groups before and after the intervention.

∗∗ P < 0.001. OLS, one-legged stance; EO, eyes opened; EC, eyes closed; FI, firm ground; FO, foam ground.

Effects on dynamic balance performance

As shown in Fig. 3, the results of the paired t tests revealed that in the intervention group, the AT, PM, and PL directions for both legs of the dynamic balance were significantly different before and after the intervention (left: AT direction: t = −3.197, P = 0.005; PM direction: t = −4.608, P < 0.001; PL direction: t = −4.706, P < 0.001; right: AT direction: t = −3.324, P = 0.004; PM direction: t = −5.614, P < 0.001; and PL direction: t = −5.905, P < 0.001) but not in the control group.

Figure 3 (A–B) Comparison of the dynamic balance performance of the participates within the groups before and after the intervention.

∗ p < 0.05, ∗∗ p < 0.001. AT, anterior; PM, posteromedial; PL, posterolateral.

Discussion

The results of this study revealed that six weeks of core stability exercises significantly improved the static balance of children and youth with IDs during the EO and FI conditions. These findings are inconsistent with those of Zhou, Zhong & Xu’s review and meta-analysis of Zhou, Zhong & Xu (2024), in which they reported that the static balance of ID participants did not significantly change compared with that of controls after the intervention. This inconsistency may be due to differences in participant characteristics and variations in the training approaches. Among the studies included in their review, only one study reported the ID severity of the participants. ID severity is considered an important factor influencing balance ability (Lipowicz et al., 2019). Additionally, the tasks used to assess static balance differed (the one-leg stance test vs. the modified stork test). Thus, the high heterogeneity caused by differences in samples and assessments might have reduced the combined effect in their meta-analysis. In addition, although the content of core stability exercises is relatively uniform, the implementation of the training process might vary among researchers.

Notably, in this study, the static balance of the participants with IDs improved only under the EO and FI conditions. This finding is understandable for the following reasons. The EO condition provided visual input to the participants with IDs, which is among the important sensory systems for maintaining balance and anticipating postural challenges (Cao et al., 2021). Visual cues provide the body with external information in the environment, which may help to compensate for the functional deficiencies of the somatosensory system and the vestibular system (Klavina, Zusa-Rodke & Galeja, 2017). Previous studies have also reported that children with IDs have greater postural instability when standing under EC conditions than under EO conditions (Lipowicz et al., 2019). Moreover, compared with the FO condition, the FI condition resulted in better static balance performance; this might be because the FI condition reduced the proprioceptive perturbation of these children. The FO condition, however, has a greater effect on the swing amplitude and weight distribution required to maintain balance; therefore, it poses a greater challenge for this population (Szafraniec, Barańska & Kuczyński, 2018). Although we found that core stability exercises affect the static balance of children and youth with IDs, this effect is not significant when visual or proprioceptive perturbations are present. Future research should consider strengthening the sensory–motor system functions in these children on the basis of core exercise. It has been reported that regularly participating in sports activities (such as soccer) can help improve proprioception and balance ability among these children (Jouira et al., 2025).

In the present study, the dynamic balance of three directions (i.e., AT, PM and PL) in children and youth with IDs greatly improved after the intervention. However, Ahmadi and colleagues (2012) revealed that after six weeks of core stability training among students with IDs, posttest results showed significant improvements in dynamic balance in the PM and PL directions, but no significant change in the AT direction. The main reason may be the difference in training protocols between the studies. In our study, Jeffreys’ core stability exercises were performed three times a week for six weeks, with each session lasting 35 min. This protocol included three levels of progressive training and was consistent with the main training principles of exercise intervention (Kasper, 2019). In addition, diverse training methods (e.g., static contraction training, dynamic training under stable conditions and unstable conditions) provide participants with comprehensive musculoskeletal and propriceptive exercises. Although the study of Ahmadi, Hasan & Hosin (2012) increased the training load every two weeks, the six-week training program was similar in content and mainly targeted the core muscles. Therefore, future research in this field should adopt a standardised protocol for core stability exercises in terms of exercise type, frequency and duration.

Furthermore, Anderson et al. (2016) suggests that core stability exercises improved the coordination of muscles involved in balance and allow for switching from a hip postural control strategy to an ankle postural control strategy. Compared with the hip strategy, the ankle strategy is simpler and more economical, and is mainly used when the human body sways due to relatively small external disturbances (Szafraniec, Barańska & Kuczyński, 2018). Biz et al. (2022) also revealed that the application of kinesiology tape provides a moderate stabilizing effect on the ankles of the athletes and has a significant effect on their postural sway. In our study, the better dynamic balance performance of children and youth with IDs might be attributed to the fact that the postural tasks (AT, PM and PL directions) that required a hip strategy before the intervention became less challenging after the intervention, and thus could be executed more economically by an ankle strategy. However, this result still needs to be verified in future studies that employ more precise assessment tools such as force platforms.

Limitations

The following limitations in the present study offer avenues for future research. First, all the participants came from a special school, which limits the generalisability of our findings. Specifically, our findings were based on a sample of children aged 11–16 years, so their applicability to younger children or older adolescents with IDs remains uncertain and warrants further investigation. Second, the lack of a follow-up assessment limits the ability to determine the long-term sustainability of the improvements observed. Therefore, implementing a more comprehensive research design with multiple measurement points is crucial in future studies.

Conclusions

In summary, six weeks of core stability exercises significantly improved the static balance of children and youth with IDs under EO and FI conditions, as well as their dynamic balance in all three directions. To further our understanding, more comprehensive studies with multiple measurement points are needed to examine the long-term effectiveness of core stability exercises on balance among this population.

Supplemental Information

Supplemental Information 1 CONSORT 2010 Checklist

Supplemental Information 2 Raw data

Translation for “ID” is “ID Balance Performance Test Data”

Supplemental Information 3 Trial protocol (Chinese)

Supplemental Information 4 Trial protocol (English)

The authors would like to thank the special schools and all participants in this study.

Additional Information and Declarations

Competing Interests

Author Contributions

Human Ethics

Clinical Trial Ethics

Data Availability

Clinical Trial Registration

The authors declare there are no competing interests.

Jing Qi conceived and designed the experiments, analyzed the data, prepared figures and/or tables, authored or reviewed drafts of the article, and approved the final draft.

Yecheng Zhong conceived and designed the experiments, performed the experiments, analyzed the data, prepared figures and/or tables, and approved the final draft.

Junjie Zhou conceived and designed the experiments, performed the experiments, prepared figures and/or tables, and approved the final draft.

Niuniu Li performed the experiments, prepared figures and/or tables, and approved the final draft.

Yuhang Han performed the experiments, prepared figures and/or tables, and approved the final draft.

Zehua Dong performed the experiments, prepared figures and/or tables, and approved the final draft.

Yan Li performed the experiments, prepared figures and/or tables, and approved the final draft.

Wenhong Xu conceived and designed the experiments, analyzed the data, prepared figures and/or tables, authored or reviewed drafts of the article, and approved the final draft.

The following information was supplied relating to ethical approvals (i.e., approving body and any reference numbers):

This research received ethical approval from the ethics committee of the Zhejiang Normal University (ZSRT2024189).

The following information was supplied relating to ethical approvals (i.e., approving body and any reference numbers):

The study was registered in the Chinese Clinical Trial Registry (ChiCTR2400088510).

The following information was supplied regarding data availability:

The raw data is available in the Supplemental File.

The following information was supplied regarding Clinical Trial registration:

ChiCTR2400088510.

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
