# Peer review of "Effects of core stability exercises on balance among Chinese children and youth with intellectual disabilities"

_PeerJ, doi:10.7717/peerj.20454_

## Round 0.1 · original submission · Major Revisions

· Academic Editor

Major Revisions

**Language Note:** The review process has identified that the English language must be improved. PeerJ can provide language editing services - please contact us at [email protected] for pricing (be sure to provide your manuscript number and title). Alternatively, you should make your own arrangements to improve the language quality and provide details in your response letter. – PeerJ Staff

·

Basic reporting

I recommend checking the English language by a native speaker.
The references are not the most relevant, many of them being old.
The Discussion section needs to be re-evaluated, adapted and reformulated by reporting on the results from the tests and the practical part.

Experimental design

Details of the procedure and tests need to be clarified to allow for the criterion of reproducing the study, to provide sufficient details for experienced and specialist readers.

Validity of the findings

The data interpretation and discussion section, but especially the conclusions, must be related to what the authors have statistically determined and what they have proposed.

Additional comments

I will attach the report that I would like the authors to study carefully.

·

Basic reporting

Firstly, I would like to commend the authors for addressing an important and underrepresented topic.

This study examines the impact of core stability exercises on balance in children and youth with intellectual disabilities through a well-structured, randomised controlled trial. The methodology is sound, and the use of both static and dynamic balance assessments—namely, the one-legged stance and Y-Balance Test—is appropriate and effectively implemented. The statistical analysis is rigorous, and the findings have clear practical relevance for professionals in special education, adapted physical activity, and rehabilitation.

The study’s use of Jeffrey’s core stability training protocol is commendable due to its progressive structure and potential for replication. Results are systematically presented, with well-organized tables and clear references to figures, even if they were not visible in the current submission.

While the manuscript is generally readable, certain sections—especially within the discussion—contain repetitive or overly lengthy sentences. Language refinement would enhance clarity and improve accessibility for international readers.

Experimental design

The study employs a valid randomised controlled design. The methodology is clearly described and includes appropriate inclusion/exclusion criteria, ethical approval, and detailed intervention procedures.

Despite these strengths, some areas would benefit from refinement. The introduction could be enhanced by integrating a stronger theoretical framework, drawing on motor learning theories, postural control models, or neurodevelopmental principles. This would provide a more solid foundation for interpreting the findings.

Validity of the findings

While dynamic balance improved across all directions, static balance showed limited gains, with only one condition (eyes open, firm surface) yielding significant improvement. This specificity should be acknowledged more explicitly in the discussion and conclusions to avoid overgeneralization.

The absence of a follow-up assessment limits the ability to determine the long-term sustainability of the improvements observed. Although not required, its acknowledgment as a limitation would increase the study’s transparency.

The discussion section would also benefit from deeper engagement with previous studies that reported inconsistent results—such as Zhou et al. (2024)—particularly about static balance. The authors could consider whether differences in sample characteristics or exercise protocols account for these discrepancies.

Additional comments

Lastly, although the study is situated in China, the potential influence of local educational or rehabilitation practices on outcomes could be briefly discussed to broaden its contextual relevance.

In conclusion, this is a well-executed study that addresses a meaningful research question using valid methods. With improvements in theoretical framing, clearer communication of findings, and minor language revisions, the manuscript will make a valuable contribution to the literature on physical activity and disability.

Reviewer 3 ·

Basic reporting

Title and Abstract
The title is ok, and the abstract is well structured, containing the main information of the study. However, in some points the abstract is repetitive and lacks clarity in grammar and flow. Please, improve it!

Key words
Please provide them in alphabetic order.

Background
The introduction is quite well structured, and it clearly develops and states the purpose of the manuscript. However, some aspects of
Line 50: Lack of training programs and limited exercise experience may also lead to motor disturbances and decreased balance capacity (Ghaeeni, Bahari & Khazaei, 2015). In addition,
muscle weakness, limited range of motion, ataxia and abnormal muscle tone are negative factors
affecting balance in this population (Jankowicz-Szymanska, Mikolajczyk & Wojtanowski, 2012).
Please, add some challenging aspects of sport activities involving children and young people with disability and quote:
https://pubmed.ncbi.nlm.nih.gov/37519377/
Tables and Figures
The number and quality of tables are appropriate to transmit the main information of the paper.

Experimental design

Methods
This section contains enough information to understand and possibly repeat the study. However, it is not well structured, and it does not reflect the Strobe Statement-Checklist for cohort studies. As it contains a lot of results, they must move into Results section.

Statistical analysis
Please provide who performed the analysis: an independent statistician or the same authors?

Validity of the findings

Results
The results presented are quite complete, reflecting the MM section, but they should be integrated as suggested previously.

Discussion
The length and content of the discussion communicates the main information of the paper. However, the authors should discuss better their results in comparison with other type of similar article.
Sometimes there are too much repetition of earlier content.
Line 265: In addition to enhancing muscle strength and the role of ankle strategies, these exercises can improve core proprioceptive ability, providing complex visual or motor environmental changes that may benefit the sensorimotor system of children and youth with intellectual disability and thereby improve their ability for balance control.
What are Anke strategies? Use of support or tapping? Please improve this aspect quoting:
https://pubmed.ncbi.nlm.nih.gov/35630037/
The authors do not recognize the limitations and strengths of the manuscript. Please provide them in a proper section.

Conclusions
The conclusions provide a clear summary of the main points of the study, and they only reflect and refer to the results of this original article.

References
The references are almost up to date, but they should be integrated as suggested previously. Please, remove those before 2010 and replaced them with newer ones.

Additional comments

Many thanks to the authors for having presented so interesting clinical research about “Effects of core stability exercises on balance in Chinese children and youth with intellectual disabilities.”
Before resubmitting the revision version of the article, please read the editorial rules carefully, and check other editorial aspects, such as text alignment, text justification at the head, etc. The language is quite good, but the manuscript should be corrected by a person of English mother tongue.

Competing interest
There are no competing interests.

Concerns
The paper does not raise any concerns (self-citations probably).

---

## Round 0.2 · accepted · Accept

· Academic Editor

Accept

Thank you for carefully addressing all the raised points in your revision. I am pleased to confirm that your article is now accepted for publication. Congratulations on your excellent work!

·

Basic reporting

The authors have responded to the observations and recommendations. The current form of the manuscript meets the requirements of "Basic reporting". Thank you

Experimental design

The research falls within the scope of the journal, the scope of the manuscript being well defined and relevant. The authors have highlighted their personal imprint, as originality and can contribute to the scope and the existing gap in this aspect, of people with special needs.
The changes made comply with the methodological and technical standard, the structure of the manuscript being clearer and more detailed.

The methods described have sufficient details and information to be replicated.

Validity of the findings

The data presented are clearer, supporting the conclusions. The conclusions are more clearly stated, are related to the original research question, and limited to the supporting results.
The conclusions are appropriately stated,

Additional comments

I have no further comments.

·

Basic reporting

The revised version shows clear improvement. Language has been professionally edited, and the text reads smoothly. The integration of motor learning theory strengthens the conceptual foundation. Figures and tables are clearly organized and well-referenced.

Experimental design

The randomized controlled design remains valid, and the intervention methods are now described more clearly. Ethical approval is complete. The addition of theoretical context improves scientific coherence, though future work could further develop postural control aspects.

Validity of the findings

The results and their interpretation are more balanced, especially regarding static vs. dynamic balance improvements. Limitations such as the absence of follow-up are properly acknowledged. Findings are consistent and well supported by the data.

Additional comments

Overall, the manuscript has been substantially improved in structure, clarity, and rigor. Only minor stylistic or formatting edits remain.

Reviewer 3 ·

Basic reporting

OK

Experimental design

OK

Validity of the findings

OK.
The authors have addressed the reviewers' comments appropriately, thereby improving the quality of the manuscript. Well done!

Additional comments

The authors have addressed the reviewers' comments appropriately, thereby improving the quality of the manuscript. Well done!